# Association between a complex community intervention and quality of health extension workers' performance to correctly classify common childhood illnesses in four regions of Ethiopia

**Theodros Getachew** [1,2]*, **Solomon Mekonnen Abebe**[2], **Mezgebu Yitayal**[2], **Lars Åke Persson**[1,3], **Della Berhanu**[1,3]

**1** Health System and Reproductive Health Research Directorate, Ethiopian Public Health Institute, Addis Ababa, Ethiopia, **2** College of Medicine and Health Sciences, Institute of Public Health, University of Gondar, Gondar, Ethiopia, **3** Department of Disease Control, London School of Hygiene & Tropical Medicine, London, United Kingdom

* tedi.getachew@yahoo.com

**Data Availability Statement:** The Ethiopian Public Health Institute (EPHI), and the London School of

## Abstract

### Background

Due to low care utilization, a complex intervention was done for two years to optimize the Ethiopian Health Extension Program. Improved quality of the integrated community case management services was an intermediate outcome of this intervention through community education and mobilization, capacity building of health workers, and strengthening of district ownership and accountability of sick child services. We evaluated the association between the intervention and the health extension workers' ability to correctly classify common childhood illnesses in four regions of Ethiopia.

### Methods

Baseline and endline assessments were done in 2016 and 2018 in intervention and comparison areas in four regions of Ethiopia. Ill children aged 2 to 59 months were mobilized to visit health posts for an assessment that was followed by re-examination. We analyzed sensitivity, specificity, and difference-in-difference of correct classification with multilevel mixed logistic regression in intervention and comparison areas at baseline and endline.

### Results

Health extensions workers' consultations with ill children were observed in intervention (n = 710) and comparison areas (n = 615). At baseline, re-examination of the children showed that in intervention areas, health extension workers' sensitivity for fever or malaria was 54%, 68% for respiratory infections, 90% for diarrheal diseases, and 34% for malnutrition. At endline, it was 40% for fever or malaria, 49% for respiratory infections, 85% for diarrheal diseases, and 48% for malnutrition. Specificity was higher (89–100%) for all childhood illnesses. Difference-in-differences was 6% for correct classification of fever or malaria

Hygiene and Tropical Medicine primarily collected the data for this manuscript. A data sharing committee has been established. This committee will review requests and provide data without any identifiers. Interested researchers may contact Mr. Atkure Defar, atid1999@yahoo.com.

**Funding:** This project is funded by Bill & Melinda Gates Foundation (OPP1132551). This research paper represents the views of the authors and does not represent the views of the funder. The funder had no role in the study design, data collection, analysis or interpretation of data.

**Competing interests:** The authors have declared that no competing interests exist.

[aOR = 1.45 95% CI: 0.81–2.60], 4% for respiratory tract infection [aOR = 1.49 95% CI: 0.81–2.74], and 5% for diarrheal diseases [aOR = 1.74 95% CI: 0.77–3.92].

## Conclusion

This study revealed that the Optimization of Health Extension Program intervention, which included training, supportive supervision, and performance reviews of health extension workers, was not associated with an improved classification of childhood illnesses by these Ethiopian primary health care workers.

## Trial registration

ISRCTN12040912, http://www.isrctn.com/ISRCTN12040912.

## Background

Respiratory infections, diarrheal diseases, and neonatal illnesses are major causes of mortality in children below the age of five years in Sub-Saharan Africa, including Ethiopia [1]. The correct classification of these illnesses is a prerequisite for further reduction of mortality [2]. However, insufficient quality of the care provided at the primary care level is a major problem in low- and middle-income countries [3]. We have previously shown that the Ethiopian health extension workers did not satisfactorily classify these common childhood diseases [4].

The integrated Community Case Management is a strategy to reduce childhood mortality and morbidity [5]. In Ethiopia, these services are delivered through a community-based health care delivery system at health posts called the Health Extension Program [6]. The programme aims at improving access to evidence-based classification, management and treatment of sick children.

In order to reach the Sustainable Development Goals by 2030, the rate of reduction in under-five mortality needs to be accelerated through rapid scale-up of effective interventions [7]. Community-based interventions have the potential to reduce morbidity and mortality in children under the age of five years [8]. Performance reviews and mentoring of primary health care workers has been effective in improving the clinical management of childhood illnesses [9]. Regular supportive supervision, in-service training, and community ownerships are also interventions that potentially could improve the performance of community health workers [10]. In addition, ascertaining necessary equipment and supplies are needed for the provision of high-quality services at the primary health care level [11].

Some studies have found that training and supervision did not improve the quality of care provided [12]. Different approaches to supervision and support to community health workers have been tried. There is no consensus regarding who should provide the supervision, where it should be done, at what frequency, with what content, and with what tools [13]. Supervision approaches should be context-specific [14]. Further operational research has been recommended to increase the understanding how to sustain improvements in performance over time [15].

Due to a low level of care seeking for childhood illnesses [16–18], the Ethiopian Ministry of Health aimed to optimize the health extension program with a complex intervention to increase the utilization of health care services for children below the age of five years. This intervention had three components: community engagement activities that aimed to increase the utilization of services, capacity building and support to the health extension workers that

intended to increase the quality of services, and activities at the district health system level, which aimed at increasing the ownership and accountability of child health services.

Although the primary outcome of this intervention was utilization of child health services, the second component included activities that potentially could improve the quality of the primary care of sick children. Hence, the objective of this study was to analyse the association between the intervention and the health extension workers' correct classification of common childhood illnesses in intervention and comparison areas at baseline and endline surveys in the four regions of Ethiopia, where the study was implemented.

## Methods

### Study design and area

The assessment was based on a pragmatic design with 26 intervention districts and 26 comparison districts with baseline and end-line surveys performed before and after the intervention. In spite of not being a proper randomized trial, the study was registered in a trial registry for transparency reasons (http://www.isrctn.com/ISRCTN12040912). The study was implemented in four regions (Amhara, Tigray, Oromia, Southern Nations, Nationalities and Peoples) of Ethiopia. The baseline survey was conducted from December 2016 to February 2017, and the endline survey was done in the same season after two years, i.e., from December 2018 to February 2019.

### Participants

A total of 200 enumeration areas had been selected to represent the districts in the four regions. A stratified cluster sampling was used to select the enumeration areas. The sampling frame was based on the population and housing census conducted for Ethiopia in 2007 [19]. Enumeration Areas (EAs) were considered as the Primary Sampling Units (PSUs) for this survey. An enumeration area had on average 150–200 households [19]. Health posts serving these areas were included in the survey. In each health post's catchment area mobilization was done to recruit sick children from 2 to 59 months to seek care at the health post. The study aimed to assess four children per health posts, who were considered ill by their caretakers and brought to the health post for assessment, disease classification, and management by the health extension workers.

### Interventions

The intervention, labelled Optimizing the Health Extension Program, was a collaborative effort to increase the utilization of child health services. To achieve this goal the project implemented three interlinked strategies, i.e., community engagement, capacity building of health extension workers, and strengthening the district health administration's ownership and accountability to the child health service program. The intervention was based on evidence generated through a rigorous barrier analysis and literature review. The intervention was implemented with support from four non-governmental organizations (PATH, UNICEF, Save the Children, and Last 10 Kilometres/John Snow Incorporated). The intervention had a preparatory phase in the second half of 2016. From the start of 2017 to October 2018, it was fully implemented, although in eight districts implementation activities were ceased between December 2017 and April 2018 for administrative reasons.

The intervention activities that composed the capacity building component, which may have an effect on quality of care, were gap-filling training on the integrated community case management of childhood illnesses, provision of registers, job aids and program toolkits,

supportive supervision of health extension workers, performance reviews and mentorship meetings for health extension workers at the district level, and training of trainers for these health workers on community-based data for decision making. The theory of change for this intervention is illustrated in a S1 Table.

### Illness classification/outcome

The outcome variable in this analysis was disease classification, measured based on the World Health Organization health facility survey tools [20] that had been pilot-tested and customized to the local context. The assessment tool comprised of an observation of the health extension workers' assessment and classification of sick children and re-examination of these children by health officers. The health officer re-examined the ill child and classified the illnesses, i.e., fever or malaria, acute respiratory tract infection, diarrhoea or dysentery, malnutrition, or ear infection according to the integrated community case management chart booklet, which displayed an integrated decision-making algorithm [21].

### Sampling

The sample size was based on the requirement that the surveys should measure changes of a fixed number of percentage points between intervention and comparison areas at baseline and endline. A sample size of 400 per group, 800 in total, would have 80% power to detect a difference of at least 15 percentage points of correct classification of illnesses as statistically significant when comparing the health extension workers' and the re-examiners' classifications.

### Assignment method

The intervention districts had been selected based on a low coverage of key maternal, newborn and child health indicators. The Regional Health Bureaus in collaboration with the study team selected similar comparison districts. Selection criteria included burden of diseases, size of the population, number of primary health care units, health service coverage data and duration of service delivery to sick children and newborns. Absence of organizations that were implementing other demand generation activities was also a selection criterion. In addition, rural districts were considered and urban administration districts were excluded, since these neither had health posts nor implementation of the integrated community case management of childhood illnesses.

### Blinding

The health extension worker uses the health post for her regular services to sick children. The health extension workers and the implementing partners were naturally not blinded to the intervention. In contrast, the field workers and health officers performing the baseline and endline surveys were blinded to the status of intervention and comparison study areas.

### Data management and analyses

Data cleaning included the checking of ranges of key variables, and checks for internal consistency. Data collection was done on Computer Assisted Field Editing, which is a data capturing approach using paper questionnaire then entering the data to a computer tablet while the data collectors were in the field using the Census and Survey Processing System (CSPro) software (US Census Bureau, Suitland, Maryland, USA). Data were transferred using Internet File Streaming System on a daily basis. Data analyses were done using STATA v14.1 (Stata Corp LP, College Station, TX, USA).

The classifications done by the health extension workers and the re-examiners were cross-tabulated to compute sensitivity and specificity [22]. Difference-in-Difference (DID) analysis was used to estimate the effect of the intervention on the classification of childhood illnesses over time between intervention and comparison groups. In this quasi-experimental design, we used baseline and endline data from intervention and comparison groups to obtain an appropriate counterfactual estimate of a plausible causal effect. Difference-in-Differences was represented by the interaction term between time and intervention group dummy variables in a Difference-in-Differences regression model [23]. Multilevel mixed logistic regression was used to analyze if differences-in-difference of correct classification of childhood illnesses were associated with the interaction term between time and intervention group in the regression model. The fixed effects, which is a measure of association, and random effects that is a measure of variation for correct disease classification, were determined by considering the clustering effect at the health posts and health extension workers as the level of variation at 0.05 significant level. In the analysis, we controlled the child age as a covariate that could affect correct classification.

$$Y = \beta 0 + \beta 1*[\text{Time}] + \beta 2*[\text{Intervention}] + \beta 3*[\text{Time}*\text{Intervention}] + \beta 4*[\text{Covariates}] + \varepsilon$$

Where

- **$Y$:** health extension workers' ability to correctly classify common childhood illnesses

- **$\beta_1$:** Average change in $y$ from the baseline to the endline time period that is common to both groups.

- **$\beta_2$:** Average difference in $y$ between the two groups that is common in both time periods.

- **$\beta_3$:** Average differential change in $y$ from the baseline to the endline time period of the intervention group relative to the comparison group.

- **$\beta_4$:** child age considered as a covariate that was controlled in the model.

In order to analyze the health extension workers' ability to correctly classify common childhood illnesses in the Difference-in-Differences analysis, we generated a new variable, coded as 1, if there was agreement between the health extension worker's and the re-examiner's classification of the child's illness, and coded as 0, if these disagreed.

## Ethical considerations

Ethical approvals were obtained from the University of Gondar (Ref O/V/P/RCS/05/371/2018), the Ethiopian Public Health Institute (Ref 613/52) and the London School of Hygiene and Tropical Medicine (Ref 16117). Information sheets of the surveys were translated into the local languages Amharic, Oromiffaa, and Tigrigna and read to caregivers to obtain written informed consent. Participants with an acute illness that could not be managed at the health post were referred to a higher-level health facility. Re-examiners informed the health extension workers of any missing diagnoses in need of treatment for immediate additional action. This trial was registered once the optimization of Health Extension program intervention had started but prior to the end of the program and evaluation.

## Results

### Participant flow diagram

The study flow chart is presented in Fig 1. Protocol deviations from study as planned was observed. Out of a total of 200 eligible health posts, six were excluded from the intervention at baseline due to security reasons and therefore not included in the endline survey. At endline,

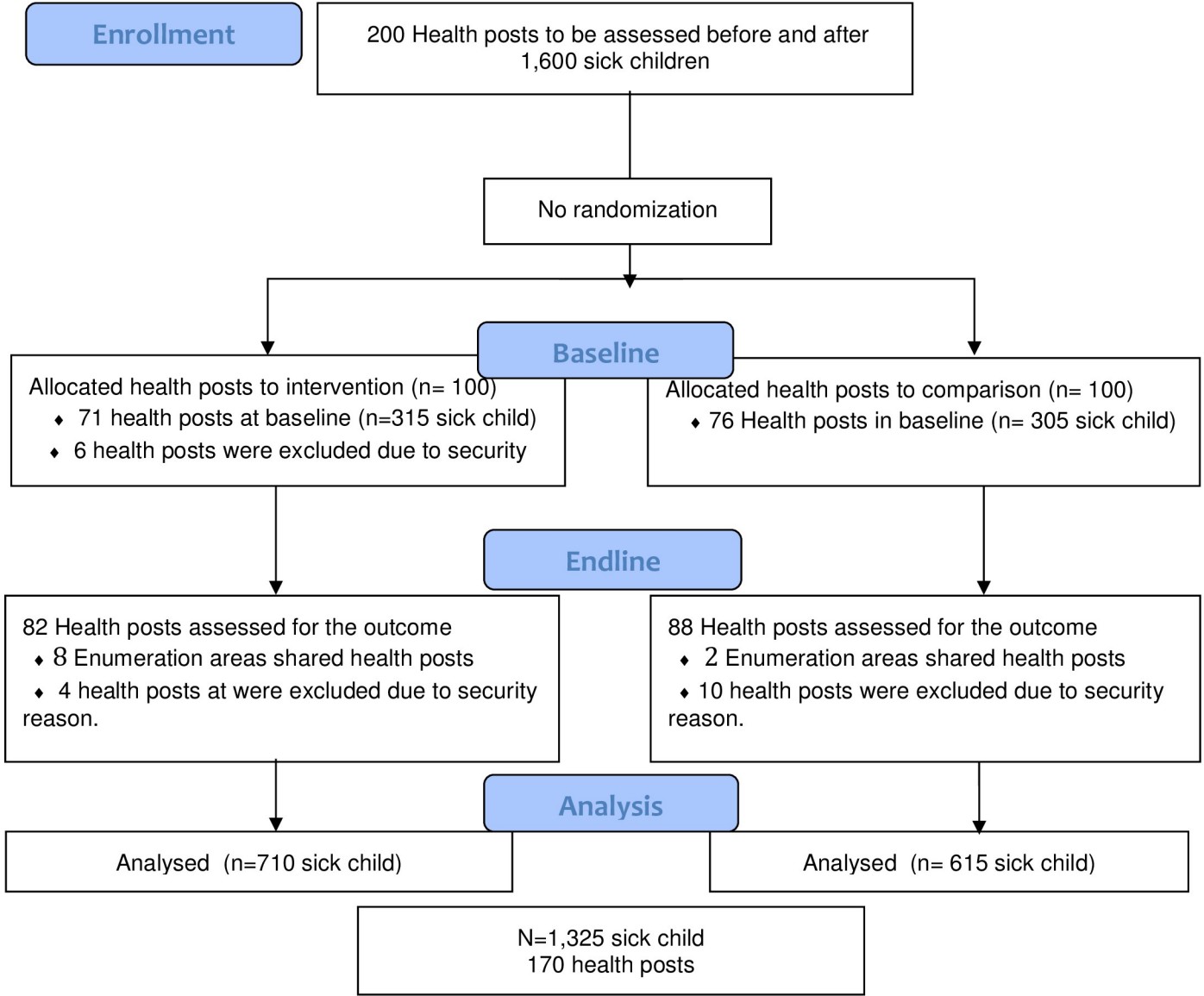

**Fig 1. Study flow diagram of intervention and comparison areas by baseline and endline data collection period.**

additional four and ten health posts were excluded in intervention and comparison areas, respectively, also due to security reasons. A total of 1,325 sick child consultations were observed and re-examined in intervention (n = 710) and comparison areas (n = 615) (Fig 1).

## Characteristics of sick under-five children

Table 1 depicts the characteristics of sick children that were assessed for common childhood illnesses in intervention and comparison areas. The characteristics of intervention and comparison districts at baseline and endline surveys were similar. A majority (75%) of the sick children in the intervention and comparison areas were from Amhara and Oromia regions. Two-thirds of the sick children were in the age group 2–23 months in the intervention as well as in the comparison areas. Among the complaints presented, cough and difficult breathing dominated in both arms.

**Table 1. Characteristics of sick 2–59 months old children at health posts who were offered integrated community case management services at baseline (December 2016- February 2017) and endline surveys (December 2018-February 2019) in intervention and comparison areas.**

| Characteristics | Baseline | | | | Endline | | | |
|---|---|---|---|---|---|---|---|---|
| | Intervention n = 315 | | Comparison n = 305 | | Intervention n = 395 | | Comparison n = 310 | |
| Region | N | % 95%CI | N | % 95%CI | N | % 95%CI | N | % 95%CI |
| Amhara | 157 | 50 (44–56) | 108 | 35 (30–41) | 183 | 46 (41–51) | 85 | 27 (23–33) |
| Oromia | 76 | 24 (20–29) | 121 | 40 (34–45) | 119 | 30 (27–35) | 151 | 49 (43–54) |
| Tigray | 50 | 16 (12–20) | 40 | 13 (10–17) | 58 | 15 (11–19) | 66 | 21 (17–26) |
| Southern Nations, Nationalities, and People | 32 | 10 (7–14) | 36 | 12 (8–16) | 35 | 9 (6–12) | 8 | 3 (1–5) |
| Age | | | | | | | | |
| 2–11 months | 123 | 39 (34–45) | 80 | 26 (21–32) | 128 | 32 (28–37) | 106 | 34 (29–40) |
| 12–23 months | 90 | 29 (24–34) | 100 | 33 (28–38) | 121 | 31 (26–35) | 89 | 29 (24–34) |
| 24–35 months | 47 | 15 (11–19) | 48 | 16 (12–20) | 66 | 17 (13–21) | 53 | 17 (13–22) |
| 36–47 months | 33 | 10 (7–14) | 42 | 14 (10–18) | 46 | 12 (9–15) | 40 | 13 (9–17) |
| 48–59 months | 22 | 7 (4–10) | 35 | 11 (8–16) | 34 | 9 (6–12) | 22 | 7 (5–11) |
| Sex | | | | | | | | |
| Boys | 168 | 53 (48–59) | 169 | 55 (50–61) | 206 | 52 (47–57) | 142 | 46 (40–52) |
| Girls | 147 | 47 (41–52) | 136 | 45 (39–50) | 189 | 48 (43–53) | 168 | 54 (48–60) |
| Complaints presented as the reason to seek care at the health post * | | | | | | | | |
| Cough or difficult breathing | 117 | 37 (32–43) | 109 | 35 (30–41) | 219 | 55 (50–60) | 170 | 55 (49–60) |
| Diarrhoea | 126 | 40 (35–46) | 98 | 32 (27–38) | 258 | 65 (60–70) | 129 | 42 (36–47) |
| Vomiting | 44 | 14 (10–18) | 40 | 13 (10–17) | 43 | 11 (8–14) | 38 | 12 (9–16) |
| Fever | 85 | 27 (22–32) | 78 | 26 (21–31) | 79 | 20 (16–24) | 58 | 19 (15–24) |
| Ear problem | 16 | 5 (3–8) | 21 | 7 (4–10) | 20 | 5 (3–8) | 13 | 4 (2–7) |

* A child may have more than one complaint; therefore, the numbers in the individual columns may add to more than the total number of observed children.

## Disease classification

Fig 2 shows the health extension workers' ability to correctly classify common childhood illnesses (sensitivity) at baseline and endline surveys. The baseline assessment indicated that the health extension workers were poor at classifying the diseases in the intervention as well as in the comparison areas, except for diarrheal diseases, which had a sensitivity of around 90% in both arms. Similarly, the health extension workers poorly classified the diseases at the endline assessment, except for diarrheal diseases that had a sensitivity of 85% (Fig 2).

Fig 3 shows the health extension workers' ability to correctly classify the healthy child (specificity) at the baseline and endline. Consequently, the baseline and endline assessment imply that these health workers were good at classifying the healthy child in the intervention as well as comparison areas (Fig 3).

Table 2 displays difference-in-differences analyses of correct classification of illnesses in intervention and comparison areas at baseline and endline surveys. Accordingly, there was no change in the correct classification of any of the childhood illnesses that was associated with the intervention.

In addition, we present the fixed effects and random effects for correct classification of childhood illnesses. The Intraclass Correlation Coefficient (ICC) estimated in the model

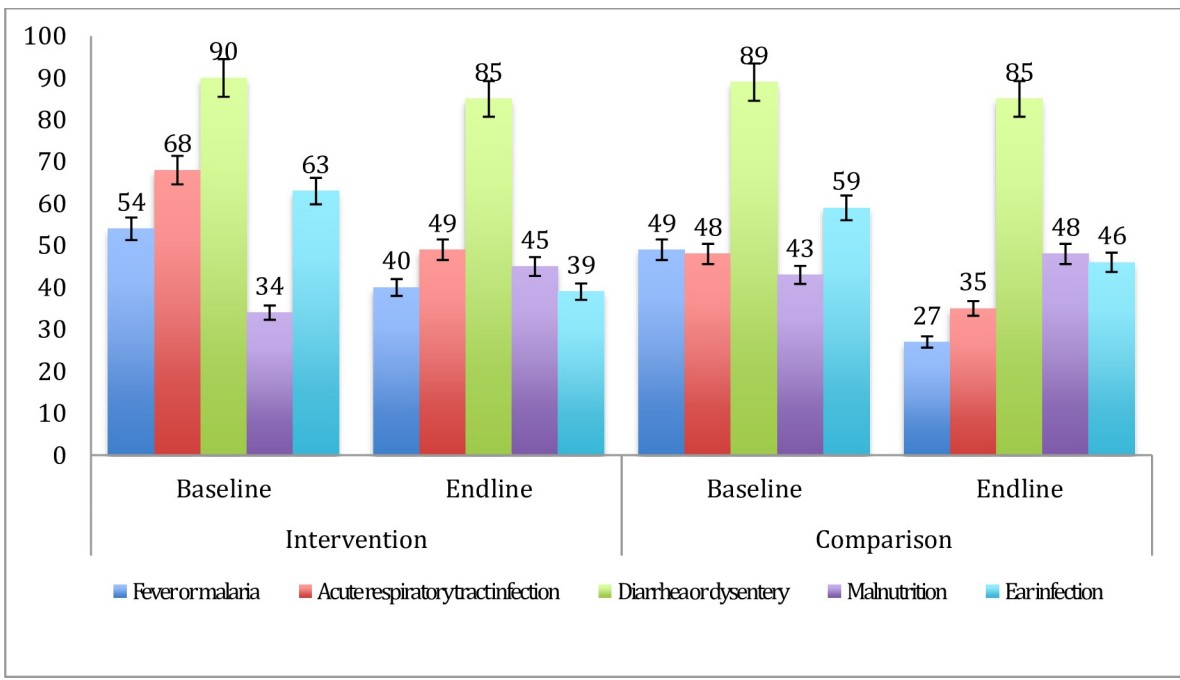

**Fig 2. Sensitivity of the health extension workers' disease classification among 2–59 months of age children at baseline (December 2016- February 2017) and endline surveys (December 2018-February 2019) in intervention and comparison areas.**

ranged from 5.4% to 16.2%, indicating that the differences in the disease classification could be attributed to the health extension workers and health posts level.

## Discussion

We found that there was no association between the Optimizing of Health Extension Program intervention and the health extension workers' ability to correctly classify sick children. The

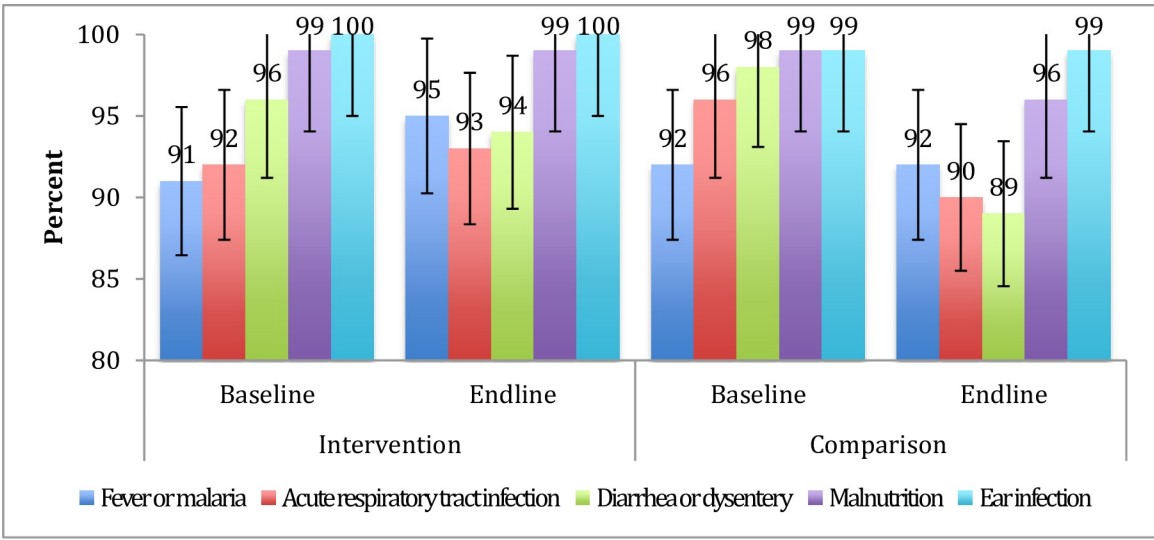

**Fig 3. Specificity of the health extension workers' disease classification among 2–59 months of age children in baseline (December 2016- February 2017) and endline surveys (December 2018-February 2019) in intervention and comparison areas.**

**Table 2. Correct classification of childhood illnesses among children 2–59 months of age in baseline (December 2016- February 2017) and endline surveys (December 2018-February 2019) in intervention and comparison areas.**

| Childhood illnesses | Baseline | | | Endline | | | | * P-value | ** ICC | *** aOR (95% CI) |
|---|---|---|---|---|---|---|---|---|---|---|
| | Intervention (n = 315) | Comparison (n = 305) | Diff | Intervention (n = 395) | Comparison (n = 310) | Diff | Difference-in-differences | | | |
| | % (95%CI) | % (95%CI) | % | % (95%CI) | % (95%CI) | % | % | | | |
| Fever or malaria | 81.7 (77.2–85.9) | 82.8 (78.6–87.3) | -1.0 | 85.8 (81.9–89.1) | 82.0 (77.2–86.0) | 3.8 | 4.1 | 0.259 | 0.162 | 1.43 (0.77–2.68) |
| Acute respiratory tract infection | 87.6 (83.5–91.0) | 88.4 (84.0–91.6) | -0.8 | 82.5 (78.4–86.1) | 77.7 (72.7–82.2) | 4.8 | 5.6 | 0.185 | 0.121 | 1.55 (0.81–2.95) |
| Diarrhea or dysentery | 93.3 (89.9–95.8) | 93.9 (91.2–96.7) | -0.6 | 90.6 (87.3–93.3) | 86.1 (81.8–89.8) | 4.5 | 5.1 | 0.203 | 0.054 | 1.71 (0.75–3.90) |
| Malnutrition | 92.1 (88.5–94.8) | 91.7 (88.9–95.2) | 0.3 | 94.7 (91.9–96.7) | 91.9 (88.3–94.7) | 2.8 | 2.5 | 0.372 | 0.111 | 1.49 (0.62–3.55) |
| Ear infection | 95.8 (93.0–97.8) | 95.9 (93.2–97.9) | -0.1 | 96.2 (93.8–97.9) | 96.8 (94.1–98.4) | -0.6 | -0.5 | 0.755 | 0.064 | 0.83 (0.276–2.65) |

* Difference-in-differences analysis was done based on multilevel mixed logistic regression.

** Multilevel mixed model was conducted to estimate the ICC.

*** The OR was adjusted for age. It represents the interaction term between time and intervention group in regression model.

poor ability to correctly classify sick children noted at the baseline survey was not changed. An exception was diarrheal diseases, which were appropriately diagnosed at baseline as well as at endline surveys. The specificity of the classification, i.e., the ability to identify healthy children, was good at both times.

The sampling of health posts included in this study was based on a pre-planned intervention activity. Thus, the sample was not selected to represent the regions, but we have no reasons to believe that these health posts and health extension workers were different from the average of these regions. The selection of intervention and comparison districts provided similar study population characteristics at baseline, except for the age category. This difference was considered in the analysis. The characteristics of the sick under-five children were also similar at baseline. The re-examiners of the sick children, the health officers, were well trained in the integrated community case management algorithms, and re-trained as part of study preparations. Still, they may theoretically have over-estimated classification of some sick children. However, a study conducted in Ethiopia indicated that the health officers had adequate and higher knowledge than midwifes and nurses in child health services [24]. Therefore, we believe that the health officers provided an adequate comparison in relation to the standard integrated community case management classification algorithm.

This study has some limitations that should be considered. The health extension workers might have altered their behaviour when being observed, a phenomenon labelled the Hawthorne effect [25, 26]. This potential bias might overestimate the performance and has been discussed in our report of the baseline findings [4]. In spite of this potential bias, a considerable proportion of sick children with common childhood illness were not appropriately classified based on their symptoms. The exclusion of some health posts due to security issues could theoretically also introduce a bias. It should, however, be noted that these facilities were few and with equal proportions excluded from both arms.

This study found no improvement of the correct disease classification linked to the intervention. In contrast, a study conducted in rural Liberia suggested significant improvements in childhood illness treatment from qualified providers after implementation of a comprehensive intervention geared to capacitate community health workers [27]. The content, intensity and

frequency of that intervention as well as the measures of outcome were different from the intervention in our study.

There is a growing awareness that quality improvements are needed to strengthen the health systems [28]. Training alone may not be sufficient to improve the performance of health care providers [3]. A majority of the health extension workers were trained in the integrated community case management of childhood illnesses that should have equipped them for the provision of good quality care to sick children.

The goal of this evaluation was to inform policymakers and underline the importance of continued efforts to improve the primary care services. In an implementation science context, a failure to achieve the targeted change can be due to failure in the implementation or deficiencies in the theory of change [29]. A recent study conducted in Tanzania showed that a multifaceted intervention, which included in-service training, mentorship, supportive supervision and infrastructure support resulted in no improvement on quality of services. Also in that study the lack of effect could be due to failure in implementation as well as failure in the suggested theory of change [30].

A recently published evaluation of the effectiveness of the Optimizing the Health Extension Program intervention showed no effect on the utilization of services for sick children [31]. The lack of effect could partly be attributed to delays, interruptions and an overall short implementation period of a complex intervention. Complex interventions that aim at behaviour changes in the health services as well as in care seeking for sick children may need an extended implementation period [32]. Previous Ethiopian studies have shown that the health extension workers were insufficiently prepared for all the tasks that fall within their scope of work [33]. This could be associated with knowledge, job satisfaction, availability of clinical guidelines, and frequency and quality of supervision [34]. A qualitative study suggested that clearly defined roles might improve the performance of the health extension workers [35]. There is currently an uncertainty about which implementation strategies work where, for whom, and under which circumstances [36]. Public health practitioners should assess the significance and feasibility of interventions in health setting prior to implementation [14]. Therefore, it is important to assess the context in which the interventions are implemented to understand the variation in effects.

## Conclusion

This evaluation in four regions of Ethiopia showed that optimizing the health extension program intervention was not associated with any positive change in the health extension workers' ability to correctly classify common childhood illnesses within the integrated community case management program. The intervention included evidence-based components that potentially could influence the quality of care provided [9, 10]. However, when combined in the upscaled intervention, there was no effect. A process evaluation to examine the quantity and quality of what was implemented showed that lack of effect could partly be attributed to delays in implementation of some initial activities that would create the condition for the successful implementation of the next activity, interruptions and an overall short implementation period of a complex intervention [32]. This could shed light on the fidelity of the implementation, the context in which this complex intervention was delivered, and explain why it did not work.

## Supporting information

**S1 Checklist. TREND statement checklist.**
(DOCX)

**S1 Table. Theory of change for OHEP interventions.**
(DOCX)

**S1 Protocol. Protocol for the evaluation of a complex intervention aiming at increased utilisation of primary child health services in Ethiopia.** A before and after study in intervention and comparison areas.
(DOCX)

## Acknowledgments

We would like to acknowledge the Institute of Public Health, University of Gondar staff members and PhD students for their constructive criticism of this manuscript during a PhD workshop.

## Author Contributions

**Conceptualization:** Theodros Getachew, Solomon Mekonnen Abebe, Mezgebu Yitayal, Lars Åke Persson, Della Berhanu.

**Data curation:** Theodros Getachew.

**Formal analysis:** Theodros Getachew.

**Funding acquisition:** Lars Åke Persson, Della Berhanu.

**Investigation:** Theodros Getachew.

**Methodology:** Theodros Getachew, Solomon Mekonnen Abebe, Mezgebu Yitayal, Lars Åke Persson, Della Berhanu.

**Project administration:** Lars Åke Persson, Della Berhanu.

**Resources:** Lars Åke Persson.

**Software:** Theodros Getachew.

**Supervision:** Solomon Mekonnen Abebe, Mezgebu Yitayal, Lars Åke Persson, Della Berhanu.

**Validation:** Solomon Mekonnen Abebe, Mezgebu Yitayal, Lars Åke Persson, Della Berhanu.

**Visualization:** Theodros Getachew.

**Writing – original draft:** Theodros Getachew.

**Writing – review & editing:** Theodros Getachew, Solomon Mekonnen Abebe, Mezgebu Yitayal, Lars Åke Persson, Della Berhanu.

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
