## [Decision Letter · Decision Letter 0]

11 Jun 2020

PONE-D-20-05286

Association between a complex community intervention and quality of health extension workers’ performance to correctly classify common childhood illnesses in four regions of Ethiopia

PLOS ONE

Dear Dr. Getachew,

Thank you for submitting your manuscript to PLOS ONE. After careful consideration, we feel that it has merit but does not fully meet PLOS ONE’s publication criteria as it currently stands. Therefore, we invite you to submit a revised version of the manuscript that addresses the points raised during the review process.

We look forward to receiving your revised manuscript.

Kind regards,

Godfrey Biemba, MBChB, M.Sc

Academic Editor

PLOS ONE

Journal Requirements:

3. Please include a caption for figure 1.

Additional Editor Comments (if provided):

The paper is well written and just requires minor revisions as recommended by Reviewer1 which I agree with. The Conclusion should include some form f a policy and/or program recommendation or perhaps a more randomized clinical trial to re-validate the negative findings or perhaps a RCT may show different results for the intervention evaluated.

Can the authors explain the absence of data on fever/malaria at endline in the comparison group in Fig 3?

Reviewers' comments:

Reviewer's Responses to Questions

**Comments to the Author**

1. Is the manuscript technically sound, and do the data support the conclusions?

Reviewer #1: Yes

2. Has the statistical analysis been performed appropriately and rigorously? 

Reviewer #1: Yes

3. Have the authors made all data underlying the findings in their manuscript fully available?

Reviewer #1: Yes

4. Is the manuscript presented in an intelligible fashion and written in standard English?

Reviewer #1: Yes

5. Review Comments to the Author

Reviewer #1: This paper is of high quality and well presented. My reservation is how useful is it to the Ethiopian health planners?While the literature quoted is good, most comments are at an overview level. Are there any specific recommendations that could be presented as to the way forward from such papers that describe successful interventions. Such as White EE, Downey J Sathananthan et al (your Ref 27) , Feyissa GT, Balabanova, Woldie M et al (your Ref 9) The latter papers (28 onwards) indicate the authors have a lot of experience in Ethiopia that suggests that specific recommendations may be useful. Also line 410, Leatherman S reference is incomplete -Journal lacking. Thanks for your work.

6. PLOS authors have the option to publish the peer review history of their article (what does this mean?). If published, this will include your full peer review and any attached files.

Reviewer #1: Yes: Paul Freeman

---

## [Author Response · Author response to Decision Letter 0]

24 Jun 2020

The map used to indicate the study area was produced by the authors using the open access reliable web site “GADM” available at https://gadm.org/download_country_v3.html. Now, we have written the source data used for production as a footnote. We have indicated the amendment using track changes in the manuscript. 

Any reasonable data request will get access to data after contacting Della Berhanu, della.berhanu@lshtm.ac.uk. The data policy of the Ethiopian Public Health Institute does not allow us to put these data to immediate access at a public repository.

A point-by-point response to the reviewers’ comments and concerns were attached with cover letter.

---

## [Decision Letter · Decision Letter 1]

8 Sep 2020

PONE-D-20-05286R1

Association between a complex community intervention and quality of health extension workers’ performance to correctly classify common childhood illnesses in four regions of Ethiopia

PLOS ONE

Dear Dr. Getachew,

Thank you for submitting your manuscript to PLOS ONE. After careful consideration, we feel that it has merit but does not fully meet PLOS ONE’s publication criteria as it currently stands. Therefore, we invite you to submit a revised version of the manuscript that addresses the points raised during the review process.

In particular, please ensure that the comments raised by reviewer 2 are addressed in full in the revised manuscript.

We look forward to receiving your revised manuscript.

Kind regards,

Emily Chenette, Deputy Editor-in-Chief, PLOS ONE

On behalf of

Godfrey Biemba, MBChB, M.Sc

Academic Editor

PLOS ONE

Reviewers' comments:

Reviewer's Responses to Questions

**Comments to the Author**

1. If the authors have adequately addressed your comments raised in a previous round of review and you feel that this manuscript is now acceptable for publication, you may indicate that here to bypass the “Comments to the Author” section, enter your conflict of interest statement in the “Confidential to Editor” section, and submit your "Accept" recommendation.

Reviewer #2: (No Response)

2. Is the manuscript technically sound, and do the data support the conclusions?

Reviewer #2: No

3. Has the statistical analysis been performed appropriately and rigorously? 

Reviewer #2: No

4. Have the authors made all data underlying the findings in their manuscript fully available?

Reviewer #2: Yes

5. Is the manuscript presented in an intelligible fashion and written in standard English?

Reviewer #2: Yes

6. Review Comments to the Author

Reviewer #2: This manuscript is really lacking in details regarding the statistical methods. Propensity score matching is proposed in the abstract and then barely touched upon in the methods. It's unclear how this improves the causal inference between the intervention groups in the article and results in a very confusing and problematic methods section.

1. (lines 107-114) I would probably combine this section with the Sampling subsection.

2. (line 109) More details on stratified cluster sampling are needed. For instance, how were strata defined? How many enumeration areas were sampled per stratum? How were EAs and health posts selected?

3. (lines 178-180) What type of regression model?

4. (lines 180-181) Sadly, there is very little information here on the propensity score methods. This is mentioned in the abstract and is implied to be a key part of your methodology, yet there is one sentence in the methods. This absolutely must be described in greater detail to satisfy #3 of the PLOS criteria for publication (https://journals.plos.org/plosone/s/criteria-for-publication). For instance, this article (https://www.sciencedirect.com/science/article/pii/S0002914903012839) is one I have used as guide, except that having p-values in table 1 is now not looked upon favorably. If you are using propensity scores, you are implying that you have covariate imbalances in between your groups. Thus, you need to demonstrate how the propensity scores have improved the balance between groups. You also need to carefully lay out your propensity score implementation. This includes methodological citations for your methods, i.e., the estimation of the propensity scores and, especially, the matching as there are many different ways to do this. You must articulate what was used to estimate the propensity scores.

5. (line 183) Each of the parameters in this equation need to be defined. I'm guessing Y is described in lines 184-187, but this is never stated.

6. (line 183) In theory, the propensity score matching should mean that no other covariates are needed in the regression model. It's unclear why covariates are needed in this final model.

7. Nowhere in the methods is it mentioned that clustering from the enumeration areas. This must be done since there may be differences between individuals sampled from one area compared to another.

8. Please indicate the level of significance for these analyses.

9. (Table 1) I don't understand why there are CIs on these numbers. They are facts about the sample and not estimated.

10. (line 253-255) This is speculative. If you want to make this statement, why not try to support it with numbers collected from a survey that was designed to be representative?

11. (lines 255-256) How do you define similar? Superiority significance testing or 95% CIs are not acceptable here. I could conclude they are different; for instance, I could conclude that 39% of the intervention sample being from 2-11 month olds is different from 26% in the comparison sample.

12. (Figure 3 and 4) Please include confidence intervals on these figures.

7. PLOS authors have the option to publish the peer review history of their article (what does this mean?). If published, this will include your full peer review and any attached files.

Reviewer #2: No

---

## [Author Response · Author response to Decision Letter 1]

22 Sep 2020

We have added details in the description of the methods used in the analyses. We have revised the management of potential confounding in the analyses. With these revisions, we consider the analyses as technically sound with data that support the conclusions. We believe that sufficient details now are provided in the methods section. We consider the statistical analyses appropriate and rigorously done.

---

## [Decision Letter · Decision Letter 2]

9 Feb 2021

Association between a complex community intervention and quality of health extension workers’ performance to correctly classify common childhood illnesses in four regions of Ethiopia

PONE-D-20-05286R2

Dear Dr. Getachew,

We’re pleased to inform you that your manuscript has been judged scientifically suitable for publication and will be formally accepted for publication once it meets all outstanding technical requirements.

Kind regards,

Sherief Ghozy, M.D., Ph.D. candidate

Academic Editor

PLOS ONE

Additional Editor Comments (optional):

Reviewers' comments:

Reviewer's Responses to Questions

**Comments to the Author**

1. If the authors have adequately addressed your comments raised in a previous round of review and you feel that this manuscript is now acceptable for publication, you may indicate that here to bypass the “Comments to the Author” section, enter your conflict of interest statement in the “Confidential to Editor” section, and submit your "Accept" recommendation.

Reviewer #2: All comments have been addressed

Reviewer #3: All comments have been addressed

2. Is the manuscript technically sound, and do the data support the conclusions?

Reviewer #2: (No Response)

Reviewer #3: Yes

3. Has the statistical analysis been performed appropriately and rigorously? 

Reviewer #2: (No Response)

Reviewer #3: Yes

4. Have the authors made all data underlying the findings in their manuscript fully available?

Reviewer #2: (No Response)

Reviewer #3: Yes

5. Is the manuscript presented in an intelligible fashion and written in standard English?

Reviewer #2: (No Response)

Reviewer #3: Yes

6. Review Comments to the Author

Reviewer #2: (No Response)

Reviewer #3: (No Response)

7. PLOS authors have the option to publish the peer review history of their article (what does this mean?). If published, this will include your full peer review and any attached files.

Reviewer #2: No

Reviewer #3: **Yes: **Abdullah Reda

---

## [Editor Report · Acceptance letter]

2 Mar 2021

PONE-D-20-05286R2 

Association between a complex community intervention and quality of health extension workers’ performance to correctly classify common childhood illnesses in four regions of Ethiopia 

Dear Dr. Getachew:

I'm pleased to inform you that your manuscript has been deemed suitable for publication in PLOS ONE. Congratulations! Your manuscript is now with our production department. 

Kind regards, 

on behalf of

Dr. Sherief Ghozy 

Academic Editor

PLOS ONE